# The Nuclear Pore Complex: A Target for NS3 Protease of Dengue and Zika Viruses

**DOI:** 10.3390/v12060583

**Published:** 2020-05-26

**Authors:** Luis Adrián De Jesús-González, Margot Cervantes-Salazar, José Manuel Reyes-Ruiz, Juan Fidel Osuna-Ramos, Carlos Noe Farfán-Morales, Selvin Noé Palacios-Rápalo, José Humberto Pérez-Olais, Carlos Daniel Cordero-Rivera, Arianna M. Hurtado-Monzón, Fernando Ruíz-Jiménez, Ana Lorena Gutiérrez-Escolano, Rosa María del Ángel

**Affiliations:** 1Department of Infectomics and Molecular Pathogenesis, Center for Research and Advanced Studies (CINVESTAV-IPN), Mexico City 07360, Mexico; luis.dejesus@cinvestav.mx (L.A.D.J.-G.); margotcervantes@yahoo.com.mx (M.C.-S.); jmreyesrz@hotmail.com (J.M.R.-R.); osram90@gmail.com (J.F.O.-R.); biologonoefarfan@gmail.com (C.N.F.-M.); selvin.palacios@cinvestav.mx (S.N.P.-R.); jholais@gmail.com (J.H.P.-O.); carlos.cordero@cinvestav.mx (C.D.C.-R.); arianna_1402@hotmail.com (A.M.H.-M.); 2Molecular Virology, School of life Sciences, The Nottingham University, Nottingham NG72QL, UK; ferrj2510@gmail.com

**Keywords:** dengue, nuclear pore complex, nucleus, zika, NS3

## Abstract

During flavivirus infection, some viral proteins move to the nucleus and cellular components are relocated from the nucleus to the cytoplasm. Thus, the integrity of the main regulator of the nuclear-cytoplasmic transport, the nuclear pore complex (NPC), was evaluated during infection with dengue virus (DENV) and Zika virus (ZIKV). We found that while during DENV infection the integrity and distribution of at least three nucleoporins (Nup), Nup153, Nup98, and Nup62 were altered, during ZIKV infection, the integrity of TPR, Nup153, and Nup98 were modified. In this work, several lines of evidence indicate that the viral serine protease NS2B3 is involved in Nups cleavage. First, the serine protease inhibitors, TLCK and Leupeptin, prevented Nup98 and Nup62 cleavage. Second, the transfection of DENV and ZIKV NS2B3 protease was sufficient to inhibit the nuclear ring recognition detected in mock-infected cells with the Mab414 antibody. Third, the mutant but not the active (WT) protease was unable to cleave Nups in transfected cells. Thus, here we describe for the first time that the NS3 protein from flavivirus plays novel functions hijacking the nuclear pore complex, the main controller of the nuclear-cytoplasmic transport.

## 1. Introduction

Viral infections transmitted by mosquitoes, for example those caused by flaviviruses such as dengue virus (DENV), yellow fever virus (YFV), West Nile virus (WNV), and Zika virus (ZIKV) represent important health challenges worldwide [1,2,3]. Flaviviruses have a positive-strand RNA genome, which encodes a polyprotein that gives rise to three structural proteins (C, E and prM) involved in the formation of viral particles and seven non-structural proteins (NS1, NS2A, NS2B, NS3, NS4A, NS4B, and NS5) essential for the replication of the viral genome [4,5,6,7]. After viral entry, the viral genome is translated and replicated in the endoplasmic reticulum (ER) [8], while viral morphogenesis takes place in the ER and the Golgi apparatus [9,10]. Every step of the flaviviral life cycle, from entry to virus release, requires close interaction with cellular proteins. Therefore, viruses hijack cellular proteins, components, and processes to be replicated in the host cell. 

Although the flaviviral replicative cycle takes place in the cytoplasm, at least three viral proteins C, NS1, and NS5 are translocated from the cytoplasm to the nucleus during infection with DENV and ZIKV in mammalian cells [11,12,13], while C, NS1, NS3, and NS5 move to the nucleus in mosquito infected cells [14]. The presence of C and NS5 proteins, as well as NS4B protein from other flaviviruses such as Japanese Encephalitis Virus (JEV) and West Nile Virus (WNV), has also been documented in the nucleus of infected cells. Although little is known about the role of these viral proteins in the nucleus, mutations that modify the nuclear localization signal (NLS) of the C and NS5 proteins and its localization in the nucleus cause a significant reduction in viral replication, suggesting that its presence in the nucleus is essential during the replicative cycle of flaviviruses [15,16,17,18,19,20]. The C protein is a highly essential protein of ~12 kDa with three nuclear localization signals (NLS) [19]. Although it has been described that the C protein transit between the nucleus and the cytoplasm, the nuclear import of the protein appears to be faster than the nuclear export.

Additionally, the C protein interacts with several nuclear and nucleolar proteins such as the death domain associated protein (DAXX), histones, nucleolin, and the heterogeneous nuclear ribonucleoprotein (hnRNP) K [21,22,23]. For NS5, it is the most abundant and most conserved flaviviral protein (105 kDa and 900 amino acids) [24]. Several NLS have been described in the sequence of the NS5 protein. Its presence in the nucleus has been related to the modulation of the IL-8 expression [25], and with the inhibition of mRNA splicing [26], supporting the idea that the presence of NS5 in the nucleus can alter the expression of specific cellular genes. 

The proteins NS1 and NS3 have been reported in the nucleus of cells infected with DENV and ZIKV. However, the specific role of both proteins in this compartment is unclear. NS3 protein with its cofactor NS2B is indispensable for the cleavage of the viral polyprotein and the replication of flavivirus since it has the function of protease and helicase [27,28,29,30]. It has been observed that ZIKV and JEV NS3 protein is located in the perinuclear region of the cell, and it associates with lamin, a component of the nuclear envelope [12,31]. This interaction in ZIKV-infected cells induces the formation of extrusion sites in the nucleus, affecting the function of the centrosome [12].

Moreover, it has been demonstrated that the nucleus in ZIKV-infected cells adopts a kidney-shaped morphology [32] where NS3 is accumulated in the concave face of these altered nuclei, supporting the idea that the presence of NS3 in the nucleus or its proximity induces changes in the function and morphology of this important organelle. Given the protease activity of NS3, this protein may be causing additional alterations in the nucleus of infected cells [32]. Specifically, the cysteine proteases with the chymotrypsin-like activity of some viruses, which replicate its genome in the cytoplasm, such as the 2A pro and 3CDpro from poliovirus (PV) and rhinovirus (HRV), alter the integrity and function of the nuclear pore complex (NPC) by the cleavage/degradation of some nucleoporins (Nup), the principal components of the nuclear pore complex (NPC). Multiple copies of 30 different Nups form the NPC, and some of these Nups participate in the bi-directional nuclear-cytoplasmic transport of proteins and RNA [33,34]. The PV and HRV proteases specifically cleave the nucleoporins Nup358, Nup214, Nup153, Nup98, and Nup62, causing essential changes in the nuclear-cytoplasmic traffic of proteins and mRNA. All these changes reduce cell translation, avoiding the antiviral immune response and favoring viral replication [35,36,37,38].

In this work, we evaluated the integrity of the NPC during infection with ZIKV and DENV. Our results show that during ZIKV and DENV infection, the integrity of Nup98, Nup153, and TPR and Nup62, Nup98, and Nup153 are altered, respectively. To determine if the NS3 of ZIKV and DENV is responsible for altering the NPC, this protease was transfected along with its NS2B cofactor into hepatocarcinoma Huh7 cells, and its subcellular localization and effect in the Nups integrity was analyzed by immunofluorescence and Western blot assays. NPC alterations were observed in ZIKV and DENV infected cells as well as in transfected cells with the NS2B3 recombinant protein, demonstrating that the NS3 protease is responsible for the Nups degradation and the NPC alteration during both flavivirus infection.

## 2. Material and Methods

### 2.1. Cell Culture and Virus

The differentiated hepatocyte derived cellular carcinoma cell line Huh-7, (kindly provided by Dr. Ana Maria Rivas, Universidad Autónoma de Nuevo León, México), was grown in advanced Dulbecco’s modified Eagle´s medium (DMEM) supplemented with 2 mM glutamine, penicillin (5 × 10^4^ U/mL) streptomycin (50 μg/mL), 10% fetal bovine serum (FBS) and 1 mL/L of amphotericin B (Fungizone) at 37 °C and a 5% CO_2_ atmosphere. Propagation of DENV serotype 2 New Guinea and serotype 4 H241 strains and ZIKV strain MEX_CIENI551 (kindly provided by Dr. Jesus Torres, Escuela Nacional de Ciencias Biológicas del Instituto Politécnico Nacional) was carried out in CD1 suckling mice brains (provided by Unidad de Producción y Experimentación de Animales de Laboratorio (UPEAL)) and focus assays determined titers in Huh-7 cells. CD1 suckling mice brain lysates from mock-infected mice were used as control. Subsequently, cells were infected with DENV and ZIKV at a multiplicity of infection MOI of 3 for 12, 24, and 48 h.

### 2.2. Focus Forming Assay

Serial dilutions of DENV or ZIKV were used to infect Huh-7 cells grown in culture plates (1 × 10^5^ cells/plate). The infection was permitted for 24 or 48 hrs. Cells were fixed with 2% formaldehyde for 20 min, treated with permeabilizing solution (1% serum, and 0.2% saponin in PBS) for 20 min at room temperature (RT) and incubated with a mouse anti-E (4G2) antibody overnight at 4 °C. Cells were incubated with anti-mouse coupled to FITC IgG, and the focus was visualized by fluorescence microscopy. Viral titer was expressed as focus forming units (FFU).

### 2.3. Transfection of Huh7 Cells

Plasmids containing the sequence of DENV and ZIKV NS2B3 protein and the inactive form of the proteases (NS2B3-S135A) were kindly donated by Dr. Ana Fernandez-Sesma and Dr. Adolfo García-Sastre (DENV) from Mount Sinai, New York, USA and Dr. Jonathan Ball (ZIKV) from the University of Nottingham, United Kingdom. These plasmids were propagated in competent *E. coli* DH5α, and purification was performed with the Zippy Plasmid Miniprep kit (ZYMO Research), following the instructions provided by the manufacturer.

Huh7 cells were transfected with plasmids at a confluence of 70%–80% using electroporation following the protocol of Hashemi et al., 2012 [39], with some modifications. Briefly, 1 × 10^7^ cells were washed with PBS and resuspended in 200 µL of OptiMem with 5 µg of DNA. The cells were transferred to a Gene Pulser cuvette with a 4mm electrode gap.

The electroporation was performed on a Gene Pulser Xcell (BioRad, Germany), electric field strength and pulse length of 170 V and 40 ms in exponential decay. Cells were cultured in advanced DMEM with 15% FBS and transfection was evaluated at 48 h.

### 2.4. Transmission Electron Microscopy

Huh-7 cells grown in p100 plates were mock infected or infected with DENV 2 or ZIKV for 24 h at an MOI of 3. We used DENV and ZIKV-infected Huh-7 cells for 24 h because in this time one cycle of flavivirus replication ends [40] (Junjhon et al., 2014). Then, the samples were fixed with 2.5% glutaraldehyde in 0.1 M sodium cacodylate buffer pH 7.2 for 1 h at room temperature (RT), and post-fixed with 1% osmium tetroxide for 1 h at RT. The samples were dehydrated through an ethanol gradient and propylene oxide, and then they were embedded in Polybed epoxy resins and polymerized at 60 °C for 24 h. Finally, 70-nm-thin sections were stained with uranyl acetate and lead citrate and, using a Jeol JEM-1011 transmission electron microscope, were analyzed (Jeol Ltd., Tokyo, Japan).

### 2.5. Immunoblotting

Infected or transfected cells were lysed with RIPA buffer (10 mM Tris-HCl pH 8, 1 mM EDTA, 0.5 mM EGTA, 1% Tritón x-100, 0.1% Desoxicolato, 0.1% SDS, and 140 mM NaCl) in the presence of protease inhibitor cocktail (ROCHE); protein extract was quantified with Pierce BCA Protein Assay Kit (Thermo Fisher Scientific) following the manufacturer’s instructions.

Cellular proteins (30–50 µg) were separated by SDS-PAGE and transferred to nitrocellulose membranes (Bio-Rad), then blocked with 10% nonfat milk in PBST (PBS-Triton X-100 0.5%) for 1 h at room temperature.

Monoclonal antibodies used for the detection of nuclear pore proteins were: rabbit polyclonal anti-Nup62 (1:6000, Abcam) and anti-Nup98 polyclonal antibodies (1:6000, Cell signalling); mouse polyclonal anti-Nup153 antibody (1:3000, Abcam); and mouse monoclonal anti-TPR antibody (1:500, Santa Cruz Biotechnology, Santa Cruz, CA). The detection of the DENV NS3, NS5, and ZIKV NS3 was performed using rabbit polyclonal antibodies (1:5000 and 1:5000, GeneTex). The anti-rabbit HRP, anti-mouse HRP, and anti-goat HRP antibodies (1:10000, Cell Signaling) were used as secondary antibodies. The proteins from the Western blotting assays were visualized with Super Signal West Femto Chemiluminescent Substrate (Thermo Scientific). Densitometric analysis was performed using the myImageAnalysis software (Thermo Fisher Scientific, Illinois, USA) and adjusted with the loading control (β-actin).

### 2.6. Confocal Microscopy

Huh-7 cells grown on slides were transfected or not with NS2B3 or NS2B3-S135A from DENV or ZIKV, or infected or not with DENV2, DENV4, or ZIKV at an MOI of 3. Cells were treated with permeabilizing solution (serum 1%, saponin 2mg/mL in PBS) for 20 min at RT. Cells were incubated with 1 µg/mL of either rabbit anti-NS5 protein, rabbit anti-NS3 protein, or mouse anti-E protein (4G2) antibodies. Nucleoporins were detected using the antibody Mab414 (Abcam) directed to the FG-Nups (Nup62, Nup58, Nup54, Nup98, Nup45, Nup214, hCG1, Nup153, and Nup50) [41], or specific anti-Nup62, anti-Nup98, anti-Nup153, or anti-TPR in permeabilizing solution overnight at 4 °C. Cells were incubated with 1 µg/mL of AlexaFluor 488-conjugated donkey anti-mouse IgG, AlexaFluor 555-conjugated goat anti-rabbit IgG, AlexaFluor 555-conjugated mouse anti-goat IgG, or AlexaFluor 488-conjugated anti-rabbit IgG. Nuclei were stained with Hoechst (Santa Cruz Biotechnology, Santa Cruz, CA) or DAPI. Slides were observed in a Zeiss LSM700 laser confocal microscope (Germany) or in a Leica TCS SP8 (Germany) as indicated. The images obtained were processed with Leica Application Suite X Core Offline software (Germany).

### 2.7. Treatment with Protease Inhibitors and Cell Viability Assay

Huh-7 cells grown in 2.3 × 10^5^ cells/plate were infected with DENV 2 at an MOI of 3 for 8 hrs. Cells were incubated with two different serine protease inhibitors (Leupeptin 1 µM or TLCK.HCl 150 µM) for 24 hrs. Cells were lysed and the integrity of the Nups was analyzed by Western blot assay using 60 μg of protein.

### 2.8. Statistical Analysis

For the statistical analysis, numerical data were expressed with means and standard deviations (SD) or standard error of the mean (SEM). The unpaired t-test and ordinary one-way ANOVA with Bonferroni’s multiple comparisons test were used and in all tests, a *p* ≤ 0.05 was considered statistically significant.

### 2.9. Ethics Statement

This study was conducted by the Official Mexican Standard Guidelines for Production, Care and Use of Laboratory Animals (NOM-062-ZOO-1999) and the protocol number 048-02 was approved by the Animal Care and Use Committee (CICUAL) at CINVESTAV-IPN, Mexico.

## 3. Results

### 3.1. DENV and ZIKV Infection Induces Changes in the Nuclear Envelope and Disrupts the Nuclear Pore Complex

Flavivirus replication occurs in the endoplasmic reticulum (ER), in close proximity to the nucleus where some viral proteins are transported [11,12,13]; for this reason, the nuclear envelope (Ne) integrity during ZIKV and DENV infection was evaluated by transmission electron microscopy (TEM). A set of invaginations of the endoplasmic reticulum (ER) membranes such as virus-induced vesicles (Ve) and vesicle packets (Vp) known as viral replication complexes [32] were observed in ZIKV- and DENV-infected Huh-7 cells (Figure 1). Virus-like particles (Vi), electron-dense small spheres with morphology and size similar to that of viral particles were detected within distended ER cisterns, confirming the ZIKV and DENV infection and the viral progeny production (Figure 1). Ultrastructural examination of the Ne from ZIKV and DENV 2 infected cells evidenced that this structure is distended and had a marked absence of electron density or variations in intensity of diffraction in the sample (diffraction contrast) in comparison with the Ne observed in mock-infected cells (Figure 1D), suggesting that ZIKV and DENV infection leads to loss of Ne integrity as it occurs in some hereditary and acquired diseases [42].

Since the NPC is an important component of the Ne, the integrity of this structure was analyzed in uninfected and infected cells. The NPC is composed of several copies of 30 different Nups; most of them are structural proteins. However, the Nups that form the central channel of the NPC harbour Phe and Gly-rich repeats (FG-Nups) that play an important role in the transport of cargo molecules through the NPC. To analyze the distribution and integrity of the Nups that are mainly involved in the nuclear-cytoplasmic transport (FG-Nups) during infection with ZIKV and DENV, the monoclonal antibody Mab414, which is directed to the FG repeats present in those Nups (Figure 2A), was used in immunofluorescence assays. Staining of a ring around the nucleus was clearly observed in mock-infected cells incubated with the Mab414 antibody; however, this ring was disassembled in cells infected with the two DENV2 and DENV4 serotypes, supporting the idea that DENV infection alters the integrity or distribution of the Nups that contain FG repeats (Figure 2A). Given the close relationship between the changes observed in the Ne in DENV- and ZIKV-infected cells, staining of the NPC with the same antibody was performed in cells infected with ZIKV. Partial disruption of the ring structure and an important reduction in the mean fluorescence intensity (MFI) of this structure was detected in cells infected with ZIKV (18.1% ± 3.5% reduction of MFI), DENV 2 (59.3% ± 4.3% reduction of MFI) and DENV 4 (45% ± 5.7% reduction of MFI), respectively, suggesting that the integrity or distribution of the Nups harbouring FG repeats are modified during infection with all these flaviviruses (Figure 2A,B).

Since the Mab414 antibody detects several Nups harboring the FG repeats, the next step was to analyze the integrity of specific Nups that are involved in RNA and protein transport through the NPC by Western blotting. Interestingly, at 24 hpi with DENV2 and DENV4, an important reduction in the amount of Nup153, Nup98, and Nup62 but not TPR was observed (Figure 2C). In contrast, at 24 hpi with ZIKV, a significant reduction in the amount of TPR, Nup153, and Nup98, but not in the amount of Nup62, was observed (Figure 2C). These results strongly suggest that DENV and ZIKV infection induced the cleavage/degradation of some Nups involved in the nuclear-cytoplasmic transport, in that Nup153 and Nup98 were altered in both DENV and ZIKV infection [43].

### 3.2. DENV and ZIKV Infection Alters the Localization and Integrity of Some Nucleoporins

The replicative cycle of flaviviruses starts with the binding and internalization of the viral particle (0 h), which results in the release of the viral nucleocapsid into the cytoplasm of the host cell (from 15 min to 4 hpi). In the cytoplasm, the viral genome is translated to produce the viral proteins in close association with the replication complexes (generated from 8 hpi) near the ER membranes. gRNA production and assembly of viral particles are carried out in the replication complexes (from 8 to 24 hpi). These viral particles are released into the extracellular space as infectious viruses from 12 hpi, thus completing one replication cycle in about 24 h [40,44]. Therefore, we decided to evaluate the integrity of Ne at 12, 24, and 48 hpi with DENV and ZIKV.

Using a specific anti-Nup62 antibody, we detected a homogenous distribution of this protein in the nucleoplasm of mock-infected cells as expected [45] (Figure 3A). However, in DENV2- and DENV4-infected cells, a significant reduction in the amount of the protein in the nucleoplasm and in MFI to DENV 2 (34.9% ± 2.5% reduction of MFI) and DENV 4 (46.2% ± 4.3% reduction of MFI) was observed at 48 hpi, without any presence in the cytoplasm (Figure 3A,B). To confirm the results, the expression levels of Nup62 during DENV and ZIKV infection were analyzed by Western blotting at different times post-infection. A slight reduction of 22% and 31% in the expression levels of Nup62 was detected at 24 hpi in cells infected with DENV2 and DENV4, respectively. The reduction in the expression levels of Nup62 at 48 hpi was more prominent (52% and 71% respectively) in both DENV serotypes (Figure 3C).

In contrast, the nuclear localization and abundance of Nup62 in ZIKV-infected cells were not altered, as observed by confocal microscopy (Figure 4A,B). In concordance with these results, the amount of Nup62 determined by Western blot assay was not modified at 12, 24, and 48 hpi in cells infected with ZIKV (Figure 4C).

The integrity and abundance of Nup98 during DENV2 and DENV4 were analyzed by confocal microscopy at 48 hrs post-infection. Using a specific anti-Nup98 antibody, the ring structure around the nucleus observed in the mock-infected cells was clearly disrupted in the cells infected with DENV. In both DENV2- and DENV4-infected cells, the signal was mainly observed in the nucleoplasm (Figure 5A) and reduction in MFI to DENV 2 (68.7% ± 5.5%) and DENV 4 (65.2% ± 4.8%) (Figure 5B), supporting the idea that an important disruption in the distribution of Nup98 occurs in infected cells. To further analyze the effect of DENV infection in the integrity of the Nup98, Western blotting was performed at 12, 24, and 48 hpi. A significant reduction in Nup98 protein levels was detected at 24 hpi with DENV2 and DENV4 (71% and 52 %, respectively) (Figure 5C). This reduction was more dramatic at 48 hpi (82% and 40 % respectively), indicating that Nup98 is cleaved/degraded at 48 hpi with DENV. Disruption of the ring structure around the nuclei was also observed in ZIKV infected cells; however, the staining observed in the nucleoplasm of DENV infected cells was less evident (Figure 5A and 6A,B). In concordance with this result, a significant reduction in Nup98 levels of 81% and 97% was observed at 24 and 48 post-ZIKV infections, respectively (Figure 6C). In summary, DENV and ZIKV induce a delocalization and cleavage/degradation of Nup98.

Since the integrity of Nups during infection of viruses from the same family do not always result in the degradation of the same nucleoporins [38,46], we wanted to determine if, in contrast to what occurs in DENV-infected cells (Figure 2B), TPR was altered during ZIKV infection. Therefore, TPR distribution and abundance were evaluated in ZIKV-infected cells. While the anti-TPR staining was observed around the nucleus and in the nucleoplasm of mock-infected cells, this staining was almost abolished in ZIKV infected cells (Figure 7A,B, upper panel). This reduction in the TPR levels was confirmed by Western blot, where up to 70% reduction in protein levels at 48 h post-ZIKV infection was obtained (Figure 7C).

The distribution and integrity of Nup153 were also determined during ZIKV infection. As for Nup98, the distribution of Nup153 was mainly around the nucleus of mock-infected cells (Figure 7A); however, in the infected cells this distribution was disrupted and a reduction of the MFI was observed (28% ± 8.8%) (Figure 7A,B lower panel), which correlates with the decreased protein levels detected by Western blot at 12, 24, and 48 hpi (Figure 7C).

### 3.3. NS2B3 Protease of DENV and ZIKV Is Responsible for Nups Cleavage/Degradation

The next question to answer was which molecule could be involved in Nups cleavage/degradation during flavivirus infection. Since DENV and ZIKV encode for a serine-protease (NS2B3), the first approach was to analyze the effect of two serine-proteases inhibitors, Leupeptin and TLCK.HCl in Nup62 and Nup98 cleavage during DENV infection (Figure 8). Both inhibitors were able to prevent Nups cleavage (Figure 8), suggesting that a serine-protease such as NS2B3 could be involved in Nup62 and Nup98 degradation. To support this possibility, cells were transfected with a plasmid encoding the NS2B3 of DENV2 and ZIKV, and the distribution, integrity and MFI of the FG-Nups were analyzed by confocal microscopy using the Mab414 antibody (Figure 9A,B). While in mock-transfected cells the Mab414 antibody stained a ring structure around the nucleus, at 24 hpt a disruption of the nuclear ring structure that was much more pronounced at 48 hpi was observed, suggesting that the NS2B3 is responsible for the FG-Nups degradation (Figure 9A,B).

To further confirm the participation of the protease activity of NS23 in the FG-Nups degradation, cells were transfected for 24 and 48 h with the DENV NS2B3-S135A mutant, which lacks the protease activity (Figure 10). The expression of this mutant protease did not alter the distribution and integrity of the FG-Nups, in comparison to the alterations observed when the WT protease was transfected, confirming that the protease activity of NS2B3 is required for the FG-Nups degradation (Figure 10A,B). Both the WT and mutant proteases were observed with a nuclear-cytoplasmic distribution at 24 hpt, while a predominantly cytoplasmic distribution was observed at 48 hpt.

To further confirm that the protease activity of NS2B3 is involved in the specific cleavage/degradation of Nup98, Nup153, and TPR during ZIKV infection, the Nups integrity was analyzed by Western blot in Huh-7 cells transfected with plasmids containing the active (ZIKV NS2-NS3B3 WT) or inactive (ZIKV NS2B3-S135A mut) sequence of the ZIKV protease. As expected, the Nup62 levels were not altered after ZIKV transfection with active (WT) or inactive (mutant) form of ZIKV NS2B3 protease, as in ZIKV-infected cells (Figure 11A,B). However, a significant reduction (*p*= < 0.05) in the amount of Nup98 was observed after the transfection with the active form of the viral protease as well as during infection (Figure 11A,B). This reduction was not observed after transfection with the inactive (mutant) form of the ZIKV NS2B3 protease (Figure 11A,B).

As observed for Nup98, the active form of the ZIKV NS2B3 (WT), as well as ZIKV infection, were able to induce a significant reduction in the Nup153 and TPR levels that was not observed in cells expressing the inactive form ZIKV NS2B3-S135A (Mut) (Figure 12), supporting the idea that ZIKV NS2B3 protease degrades directly or indirectly Nup98, Nup153, and TPR while DENV NS2B3 is responsible for the direct or indirect cleavage/degradation of Nup62, Nup98, and Nup153.

## 4. Discussion

During the flavivirus replicative cycle, some viral proteins are relocated to the nucleus. Specifically, these are the C protein that participates in viral assembly and encapsidation and the NS5 that is an RNA-dependent RNA polymerase with a methyltransferase activity [14,16]. In addition, several nuclear proteins have also been observed to relocate from the nucleus to the cytoplasm during flaviviral infections. While some of the nuclear proteins that relocate to the cytoplasm are degraded (such as DDX21 and DDX56), La, PTB and some others are necessary for an efficient viral replication or to inhibit the innate immune response [21,22,47,48,49,50,51].

Considering that viruses can hijack important components of the host cell, we analyzed possible alterations in the structure of Ne and the NPC, the primary regulator of the nuclear-cytoplasmic transport. Previous reports have determined affectations in the morphology of the nucleus and nuclear envelope during ZIKV and DENV infection. These studies described the formation of extrusion sites in the nucleus that may affect the function of the centrosome and nuclear lamina during ZIKV infection [12]. On the other hand, the formation of strand-like structures in mosquito cell nuclei during DENV infection have been observed [14]. Thus, it was possible that other components of the NE are also altered during infection. We analyzed the nuclear envelope during infection with ZIKV, using electron microscopy. Ultrastructural examination of ZIKV-infected cells showed changes in nuclear morphology where Ne is distended and has a marked absence of electron density in contrast to the Ne of imitation mock-infected cells.

Abnormalities in the nuclear envelope during degenerative diseases, cancer, laminopathies, and viral infections generate cellular stress such as erosion, in which NPC components are deregulated. In addition to Ne rupture, together, these abnormalities affect the bidirectional transport between the nucleus and the cytoplasm through the NPC [42,52,53,54].

The integrity of NPC in ZIKV- and DENV-infected cells was analyzed by confocal microscopy using the monoclonal antibody Mab414 that recognizes FG (Phe-Gly)-contained in FG-Nups (Nup62, Nup58, Nup54, Nup98, Nup45, Nup214, hCG1, Nup153, and Nup50) [41]. Affectations in the distribution of FG-Nups were observed in the cells infected with ZIKV and DENV infection, in which a cytoplasmatic distribution is observed, contrary to the mock-infected cells in which the distribution of FG-Nups is perinuclear. These observations indicate an alteration of the NPC during infection with ZIKV and DENV. Alterations in nucleoporin distribution are also observed during infection with DENV2 and HCV, where Nup98 is relocated to the cytoplasm [55].

Our results indicate that during DENV infection, the integrity and distribution of at least Nup153, Nup98, and Nup62 were disrupted, while during ZIKV infection, the integrity of TPR, Nup153, and Nup98 were altered. Differential target nucleoporins by the NS2B3 from DENV and ZIKV could be due to structural differences between both proteases, since they only have 67% homology in their sequence. This differencial targeting of NUPs has also been observed with different serotypes of rhinovirus and also between different picornaviruses [38,56,57].

In this work, several lines of evidence indicate that the viral protease NS2B3 is involved in Nups cleavage. First, the serine-protease inhibitors, TLCK and Leupeptin, prevented Nup98 and Nup62 cleavage. Interestingly, the cleavage products of Nup 98 (cp 1) and Nup 62 (cp 1 and cp 2) during DENV infection (Figure 8) were not observed in Huh-7 cells transfected with NS2B3 protease of ZIKV (Figure 11 and Figure 12) which could be due to their structural differences, a phenomenon observed with rhinovirus 2A proteases of different species and serotypes [38,56,57].

Second, the transfection of DENV and ZIKV NS2B3 protease was sufficient to inhibit the nuclear ring recognition detected in mock-infected cells with the Mab414 antibody. Third, the mutant but not the WT protease was unable to cleave Nups in transfected cells. Of interest, the location of the viral protease was observed in the nuclei as well as in the perinuclear region, where the replicative complexes are located. Degradation of nucleoporins by the activity of viral proteases located in both the nuclei and the perinuclear region, such as the 3CD and the 2A proteases from PV has been reported [58,59], suggesting that NS3 could be located in the perinuclear region or the nuclei could reach and cleave its target Nups. The fact that both recombinant WT proteases from DENV and ZIKV but not the recombinant mutant proteases had the ability to degrade several of these Nups strongly indicates that they are targets of these active viral proteases. However, we cannot rule out the possibility that a cellular serine protease activated by the viral NS2B3 activity could be responsible for Nups degradation. Either directly or indirectly, the viral proteases are responsible for the Nups processing.

Up to now, there is evidence generated from our group that demonstrates that NS3 reaches the nucleus in mosquito cells infected with DENV [14]. NS3 proteins from the four DENV serotypes and ZIKV contain both a putative nuclear localization sequence (NLS) and a nuclear export sequence (NES) between amino acids 270 to 282 of the NS3 protein. The presence of both sequences within the sequence of NS3 suggests that NS3 is able to shuttle between the nucleus and the cytoplasm during infection. The role of these NLS and NES in NS3 proteases import and export to the nuclei in transfected cells is currently being investigated.

This is the first report to show evidence of the effect of flavivirus infection in this important nuclear structure. Some other cytoplasmic viruses utilize the NPC as an attractive anti-host target. Picornaviruses are particularly expert in disturbing the NPC; for example, the amino-terminal Leader protein (L) from the encephalomyocarditis virus (EMCV) induces hyperphosphorylation of Nup62, Nup153, and TPR, through a Ran-dependent, mitogen-activated protein kinase cascade [60], disrupting their activity, while the viral protease 2Apro of PV and HRV cleaves Nup62, Nup98, and Nup153 [46,61,62], disrupting NPC activity. The Nups degraded during DENV and ZIKV infection play important roles in mRNA export as well as in protein transport. As with other processes, flavivirus hijacks the main regulator of the nuclear-cytoplasmic transport, which is the NPC; thus, it is possible that this cleavage/degradation of Nups causes a drastic inhibition in mRNA export or in nuclear protein transport. Considering this last process, many signaling pathways end with the translocation of transcription factors to the nucleus to induce anti-viral gene expression; under infection conditions, even though the transcription factors could be activated, their transport into the nucleus may be inhibited. For example, alterations in the nuclear transport of transcription factors such as IRF3 and NFκB inhibit the antiviral response via Interferon β1 due to the processing of Importin β1 by NS3/4A of HCV (Family Flaviviridae) [63]. For this reason, further studies are required to determine the effect of Nups cleavage/degradation in the nuclear-cytoplasmic transport of transcription factors and viral replication [64].

Analyzing the time of infection in which Nups degradation occurred, we found that Nup98 is the first NUP cleaved (from 12 h for ZIKV and at 24 h for DENV). Interestingly, it has been described that Nup98 is also a transcription factor involved in the expression of immune response genes [65,66]. This would suggest that its degradation alters the expression of Nup98-dependent genes such as CDK9, RNAPII, and HLA-DRA, which in turn regulate the expression of TNF-alpha, inducible IL-6, and IFN-gamma [65]. All these cytokines are relevant during flavivirus infection [67], and its abundance has to be evaluated after transfection with WT and mutant viral proteases.

In summary, our results describe for the first time that infection with flaviviruses such as DENV and ZIKV induces the disruption and degradation of some FG-Nups by the activity of NS2B3 protease. It is likely that this function of NS3 has a significant impact in promoting the presence of nuclear protein in the cytoplasm of the infected cells as well as inhibiting the export of mRNAs, having a role for both an efficient viral replicative cycle as well as in the modulation of the immune response. However, more studies are needed to confirm this involvement in the participation of viral replication and how this effect favors viral replication.

## Figures and Tables

**Figure 1 viruses-12-00583-f001:**
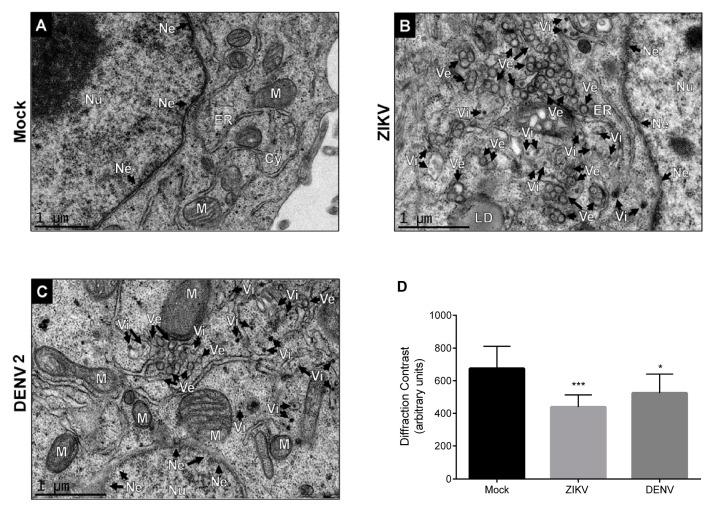
Zika virus (ZIKV) infection induces challenge in envelope nuclear in Huh-7 cells. Transmission electron microscopy of (**A**) mock-infected cells or (**B**) infected cells with ZIKV and (**C**) dengue virus (DENV 2) for 24 h. (**D**) The graphs represent the results of diffraction contrast expressed as arbitrary units and the error bars show the SD of three independent experiments. * *p* ≤ 0.05; *** *p* ≤ 0.001, *n* = 10 per group. Ne, nuclear envelope; Nu, nucleus; ER, endoplasmic reticulum; VE, double-membrane vesicles; LD, lipid droplets.

**Figure 2 viruses-12-00583-f002:**
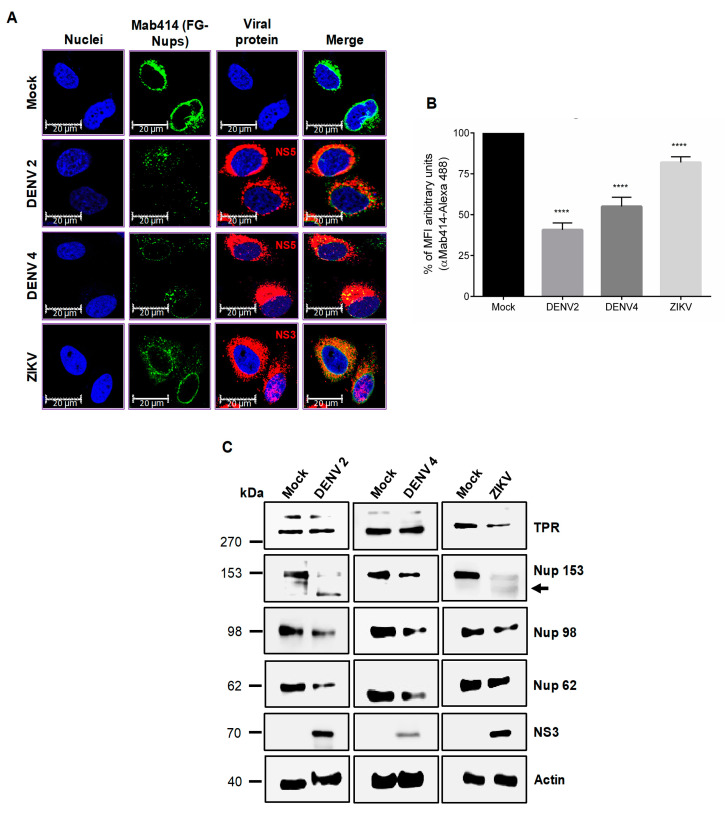
The integrity of the nuclear pore complex is altered after DENV and ZIKV infection. (**A**) Huh-7 cells mock-infected (Mock) or infected with DENV2, DENV4, or ZIKV for 48 hrs, were incubated with the monoclonal antibody against the FG-rich sequence of nucleoporins, Mab414, and analyzed by confocal microscopy. Anti-ZIKV-NS3 and anti-DENV-NS5 antibodies were used as controls of infection. Nuclei were stained with Hoechst. Representative images of three independent experiments are presented. (**B**) The graphs represent the results expressed as a percentage of Mock-normalized mean fluorescence intensity (MFI) arbitrary units and the error bars show the SD of three independent experiments. **** *p* ≤ 0.0001, *n* = 30 per group. (**C**) Levels of TPR, Nup153, Nup98, and Nup62 proteins were analyzed by Western blot in whole-cell lysates obtained from mock- and DENV2-, DENV4-, and ZIKV-infected cells for 24 hrs. The anti-NS3 antibody was used as a control of infection and anti-actin as a loading control. Representative Western blot assays of three independent experiments are presented.

**Figure 3 viruses-12-00583-f003:**
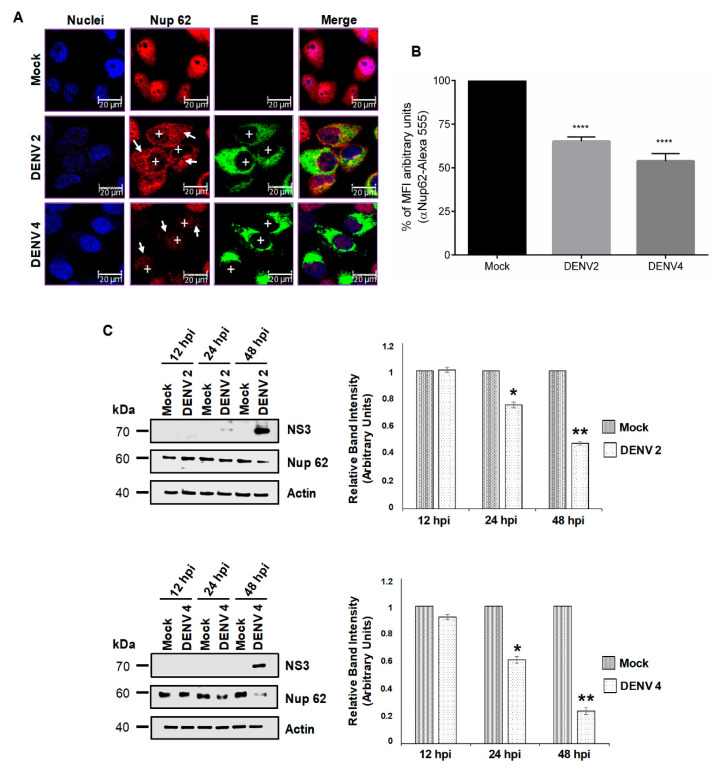
The integrity and location of the Nup62 are altered during DENV infection. (**A**) Huh-7 cells mock-infected (Mock) or infected with DENV2 or DENV4 for 48 hrs were incubated with a monoclonal anti-Nup62 antibody and analyzed by confocal microscopy. The anti-E protein antibody was used as control of infection. Nuclei were stained with Hoechst. Representative images of three independent experiments are presented. White boxes represent zoom areas. Arrows indicate non-infected cells, and plus signs indicate infected cells. (**B**) The graphs represent the results expressed as a percentage of Mock-normalized mean fluorescence intensity (MFI) arbitrary units and the error bars show the SD of three independent experiments. **** *p* = ≤ 0.0001, *n* = 30 per group. (**C**) Levels of Nup62 protein were analyzed by Western blot in whole-cell lysates from mock or DENV2 or DENV4, infected cells for 12, 24, and 48 hrs. The anti-NS3 antibody was used as a control of infection and anti-actin as a loading control. Representative Western blot assays from three independent experiments are presented. The graph represents Nup62 protein levels compared with β-actin. Protein amount values from control (mock-infected) were adjusted to a value of 1. Values for expression in infected cells (white) were then expressed as a number relative to the control. Data are means and the error bars show the SEM of *n* = 3 independent experiments performed by duplicate. * *p* < 0.05, ** *p* < 0.001.

**Figure 4 viruses-12-00583-f004:**
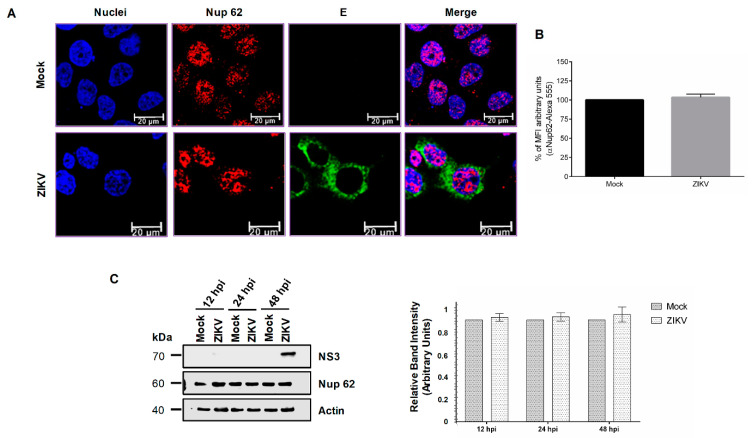
The integrity and location of the Nup62 are altered during ZIKV infection. (**A**) Huh-7 cells mock-infected (Mock) or infected with ZIKV for 48 hrs, were incubated with a monoclonal anti-Nup62 antibody and analyzed by confocal microscopy. The anti-E protein antibody was used as control of infection. Nuclei were stained with DAPI. Representative images of three independent experiments are presented. (**B**) The graphs represent the results expressed as a percentage of Mock-normalized mean fluorescence intensity (MFI) arbitrary units and the error bars show the SD of three independent experiments, *n* = 30 per group. (**C**) Levels of Nup62 protein were analyzed by Western blot in whole-cell lysates from mock or ZIKV infected cells for 12, 24, and 48 hrs. The anti-NS3 antibody was used as a control of infection and anti-actin as a loading control. Representative Western blot assays from three independent experiments are presented. The graph represents Nup62 protein levels compared with β-actin. Protein amount values from control (mock-infected) were adjusted to a value of 1. Values for expression in infected cells (white) were then expressed as a number relative to the control. Data are means ± SEM of *n* = 3 independent experiments performed by duplicate.

**Figure 5 viruses-12-00583-f005:**
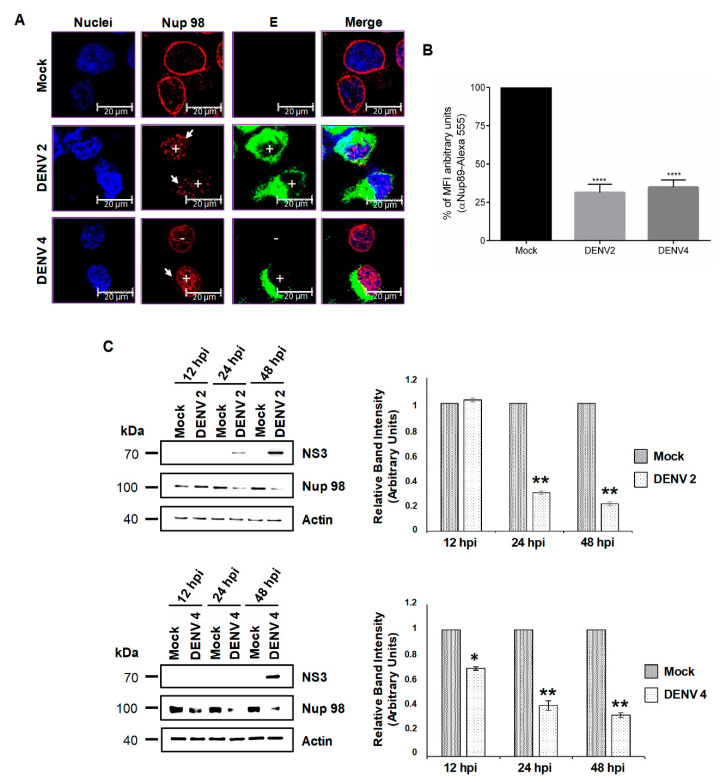
The integrity and location of the Nup98 are altered during DENV infection. (**A**) Huh-7 cells mock-infected (Mock) or infected with DENV2 or DENV4 for 48 hrs were incubated with a monoclonal anti-Nup98 antibody and analyzed by confocal microscopy. The anti-E protein antibody was used as control of infection. Nuclei were stained with Hoechst. Representative images of three independent experiments are presented. White boxes represent zoom areas. Arrows indicate non-infected cells, and plus signs indicate infected cells. (**B**) The graphs represent the results expressed as a percentage of Mock-normalized mean fluorescence intensity (MFI) arbitrary units and the error bars show the SD of three independent experiments. **** *p* ≤ 0.0001, *n* = 30 per group. (**C**) Levels of Nup-98 protein were analyzed by Western blot in whole-cell lysates from mock or DENV2, DENV4, and ZIKV-infected cells for 12, 24, and 48 hrs. The anti-NS3 antibody was used as a control of infection and anti-actin as a loading control. Representative Western blot assays of three independent experiments are presented. The graph represents Nup98 levels comparing with β-actin. Protein amount values from control (mock-infected) were adjusted to a value of 1. Values for expression in infected cells (white) were then expressed as a number relative to the control. Data are means ± standard error (S.E) of *n* = 3 independent experiments performed by duplicate. * *p* < 0.05, ** *p* < 0.001.

**Figure 6 viruses-12-00583-f006:**
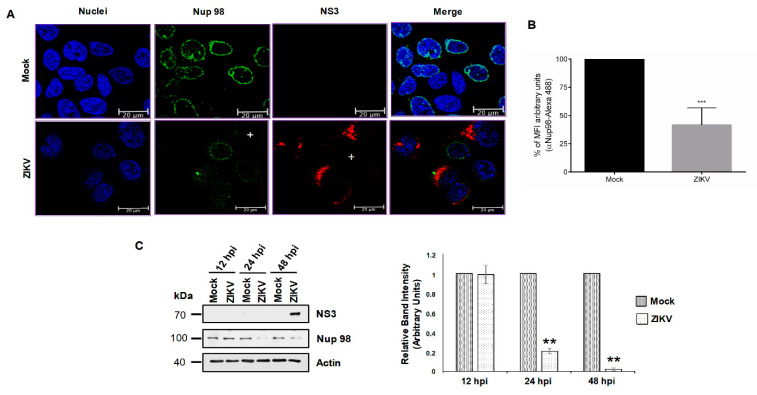
The integrity and location of the Nup98 are altered during ZIKV infection. (**A**) Huh-7 cells mock-infected (Mock) or infected with ZIKV for 48 hrs were incubated with a monoclonal anti-Nup98 antibody and analyzed by confocal microscopy. The anti-E protein antibody was used as control of infection. Nuclei were stained with DAPI. Representative images of three independent experiments are presented. Plus signs indicate infected cells. (**B**) The graphs represent the results expressed as a percentage of Mock-normalized mean fluorescence intensity (MFI) arbitrary units and the error bars show the SD of three independent experiments. *** *p* ≤ 0.001, *n* = 30 per group. (**C**) Levels of Nup98 protein were analyzed by Western blot in whole-cell lysates from mock or ZIKV-infected cells for 12, 24, and 48 hrs. The anti-NS3 antibody was used as a control of infection and anti-actin as a loading control. Representative Western blot assays of three independent experiments are presented. The graph represents Nup98 levels comparing with β-actin. Protein amount values from control (mock-infected) were adjusted to a value of 1. Values for expression in infected cells (white) were then expressed as a number relative to the control. Data are means ± standard error (S.E) of *n* = 3 independent experiments performed by duplicate. ** *p* < 0.001.

**Figure 7 viruses-12-00583-f007:**
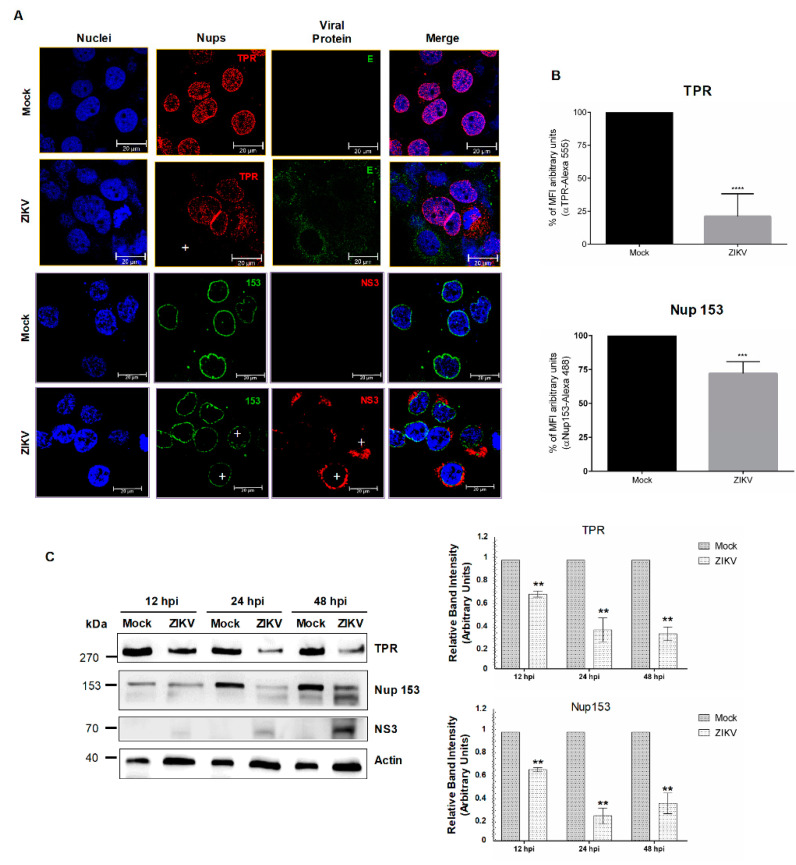
The integrity and location of TPR and Nup153 are altered during ZIKV infection.(**A**) Huh-7 cells mock-infected (Mock) or infected with ZIKV for 48 hrs were incubated with a monoclonal anti-TPR or anti-Nup153 antibody and analyzed by confocal microscopy. The anti-E protein antibody or anti-NS3 protein was used as control of infection. Nuclei were stained with DAPI. Representative images of three independent experiments are presented. Plus signs indicate infected cells. (**B**) The graphs represent the results expressed as a percentage of Mock-normalized mean fluorescence intensity (MFI) arbitrary units and the error bars show the SD of three independent experiments. **** *p* ≤ 0.0001, *n* = 30 per group. (**C**) Levels of TPR and Nup-153 proteins were analyzed by Western blot in whole-cell lysates from mock- or ZIKV-infected cells for 12, 24, and 48 hrs. The anti-NS3 antibody was used as a control of infection and anti-actin as a loading control. Representative Western blot assays of three independent experiments are presented. The graph represents TPR and Nup-153 levels comparing with β-actin. Protein amount values from control (mock-infected) were adjusted to a value of 1. Values for expression in infected cells (white) were then expressed as a number relative to the control. Data are means ± standard error (S.E) of *n* = 3 independent experiments performed by duplicate. ** *p* < 0.001.

**Figure 8 viruses-12-00583-f008:**
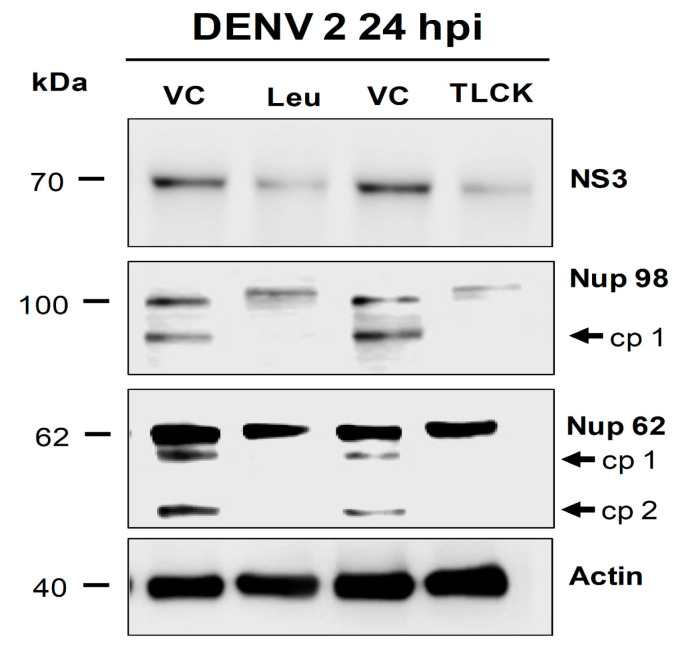
Nup98 and Nup62 are cleaved by a serine-protease during DEN infection. Huh-7 cells infected with DENV2 for 24 hrs were incubated in the absence (VC) or in the presence of the serine-proteases inhibitors Leupeptin (Leu) and TLCK. The integrity of Nup98 and Nup62 was analyzed by Western blot in whole-cell lysates. The anti-NS3 antibody was used as a control of infection and anti-actin as a loading control. Representative Western blot assays of three independent experiments are presented. Cleavage products from serine protease processing (cp) are shown.

**Figure 9 viruses-12-00583-f009:**
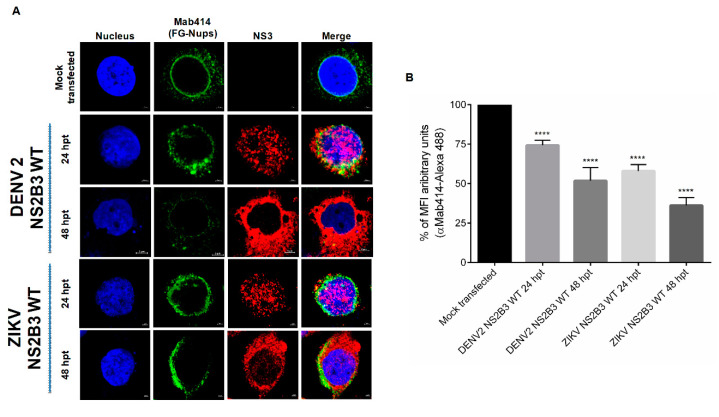
Transfection of NS2B3 proteins from DENV2 and ZIKV induces disruption of FG-rich sequence nucleoporins. (**A**) Huh-7 cells mock-transfected or transfected for 24 and 48 hrs with DENV2 or ZIKV active (WT) NS2B3 proteins were incubated with the Mab-414 and the anti-NS3 protein antibody respectively, and the integrity and subcellular localization of NS3 protein were analyzed by confocal microscopy. Nuclei were stained with Hoechst. Representative images of three independent experiments are presented. (**B**) The graphs represent the results expressed as a percentage of Mock-normalized mean fluorescence intensity (MFI) arbitrary units and the error bars show the SD of three independent experiments. **** *p* ≤ 0.0001, *n* = 30 per group.

**Figure 10 viruses-12-00583-f010:**
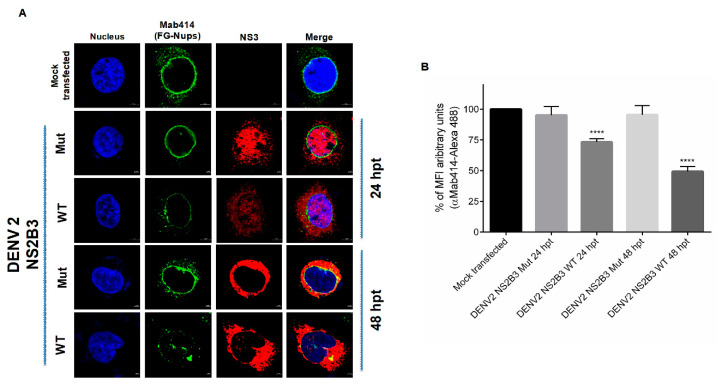
Transfection of the wild type but not with the mutant NS2B3 protein from DENV2 induces disruption of FG-rich sequence nucleoporins. (**A**) Huh-7 cells transfected for 24 and 48 hrs with wild type and DENV2 mutant NS2B3 protein. Nucleoporin integrity and subcellular localization of NS3 were analyzed by confocal microscopy using the Mab-414 and the anti-NS3 protein antibody, respectively. Nuclei were stained with Hoechst. Representative images of three independent experiments are presented. (**B**) The graphs represent the results expressed as a percentage of Mock-normalized mean fluorescence intensity (MFI) arbitrary units and the error bars show the SD of three independent experiments. **** *p* ≤ 0.0001, *n* = 30 per group.

**Figure 11 viruses-12-00583-f011:**
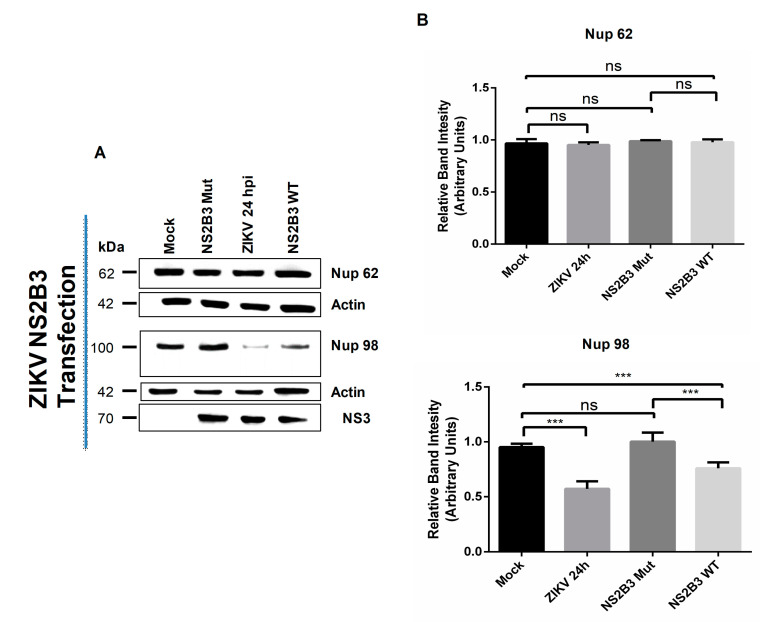
The active NS2B3 protease of ZIKV alters the integrity of Nup98, but not the Nup-62. (**A**) Huh-7 cells mock-infected (Mock) or infected with ZIKV or transfected with the active (WT) or inactive (mutant) form of the NS2B3 protease for 48 hrs were incubated with a monoclonal anti-Nup62 or anti-Nup98 antibody and analyzed by Western blot. The anti-NS3 antibody was used as a control of infection and anti-actin as a loading control. Representative Western blot assays of three independent experiments are presented. (**B**) The graph represents Nup62 and Nup98 levels comparing with β-actin. Protein amount values from control (mock-infected) were adjusted to a value of 1. Values for expression in infected cells (white) were then expressed as a number relative to the control. Data are means ± standard error (S.E) of *n* = 3 independent experiments performed by duplicate. *** *p* ≤ 0.001, ns: no significant difference.

**Figure 12 viruses-12-00583-f012:**
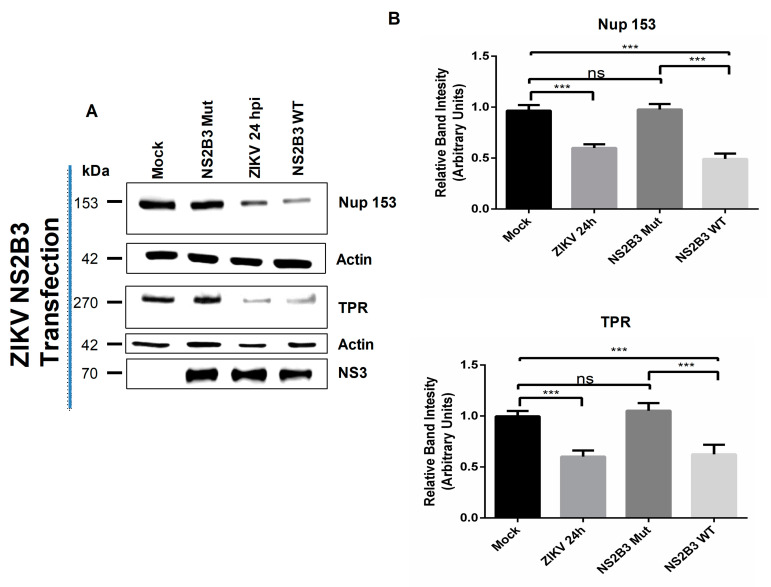
The active NS2B3 protease of ZIKV alters the integrity of TPR and NUp153. (**A**) Huh-7 cells mock-infected (Mock) or infected with ZIKV or transfected with the active (WT) or inactive (mutant) form of the NS2B3 protease for 48 hrs were incubated with a monoclonal anti-TPR or anti-Nup153 antibody and analyzed by Western blot. The anti-NS3 antibody was used as a control of infection and anti-actin as a loading control. Representative Western blot assays of three independent experiments are presented. (**B**) The graph represents Nup153 and TPR levels comparing with β-actin. Protein amount values from control (mock-infected) were adjusted to a value of 1. Values for expression in infected cells (white) were then expressed as a number relative to the control. Data are means ± standard error (S.E) of *n = 3* independent experiments performed by duplicate. * *p* < 0.05, **** p* ≤ 0.001, ns: no significant difference..

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
