# Peer review of "The Nuclear Pore Complex: A Target for NS3 Protease of Dengue and Zika Viruses"

_viruses, 2020, doi:10.3390/v12060583_

Round 1

Reviewer 1 Report

This report examines the impact of flaviviruses on the nuclear envelope and nuclear pore complex (NPC). The study utilizes ZIKV (MEX_CIENI551) and two strains of DENV (2 New Guinea and 4 H241) in Huh-7 cells. Transmission electron microscopy shows a loss of electron density at the NE of cells infected with both viruses. Confocal microscopy using an antibody that reacts with a subset of FG-containing NPC proteins (Nups) reveals loss of staining in infected cells. Immunoblot using antibodies specific for a variety of Nups reveals that both viruses proteolyze Nup98 and Nup153. Interestingly, ZIKA targets Tpr but not Nup62, while DENV targets Nup62 but not Tpr. More detailed analysis, again using Nup-specific antibodies reveals that these cleavage events result in loss of staining at the nuclear envelope with cleavage apparent by about 24 hpi, althouth Tpr and Nup153 cleavage in ZIKV-infected cells is apparent by 12 hpi. The addition of serine protease inhibitors prevents cleavage during DENV infection and expression of NS2B-NS3 via transient transfection recapitulates cleavage of Nups.

General Comments:

This study presents a large amount of data that supports the author’s conclusions that both ZIKV and DENV proteolyze NUPS via their NS2B-NS3 protease. These results are significant and convincing. The authors should consider adding additional statistical analyses to support their microscopy data such as the percentage of examined cells that display the phenotype presented in the figures. Another question is if the cleavage products produced by overexpression of the NS2B-NS3 protease mimic those found in infected cells. A minor point, but it isn’t clear why an analysis of Nup153 and Tpr in DENV-infected cells wasn’t undertaken, given that all the other virus/nup combinations were examined so thoroughly. Lastly, it would be helpful for the reader who isn’t a flavivirus expert if the authors include some background on the time course of these infections. What do 12, 24 and 48 hpi correspond to in terms of the viral life cycle.

Specific Comments:

Fig1. Details on how the TEM was done should be added to the materials and methods and/or figure legend. This should include moi.

Fig 2. Statistical analysis should be added her showing the percentage of infected cells showing reduced 414 staining. Why the difference in time for microscopy vs western?

Fig 3- how do the authors explain the lack of NE staining with the Nup62 ab? This is very different from what was seen with mAb414, which reacts with Nup62 and raises questions about the staining conditions and possible artifacts. The percent of cells showing loss of nuclear stain should be shown. If it’s similar to results obtained with mAb414 this would support the validity of the staining with this Ab.

Figure 5,6- Again, statistics showing the % of cells showing the give pattern should be added.

Fig 7- For Nup153 the loss of staining is not as apparent. Also some cells pos for viral E protein, seem to show normal staining of Nup153. Again, statistics may support the authors claims.

Fig 9/10- statistics again are needed here to support the author’s interpretation

Author Response

Reviewer 1

Thank you very much for your comments.

This report examines the impact of flaviviruses on the nuclear envelope and nuclear pore complex (NPC). The study utilizes ZIKV (MEX_CIENI551) and two strains of DENV (2 New Guinea and 4 H241) in Huh-7 cells. Transmission electron microscopy shows a loss of electron density at the NE of cells infected with both viruses. Confocal microscopy using an antibody that reacts with a subset of FG-containing NPC proteins (Nups) reveals loss of staining in infected cells. Immunoblot using antibodies specific for a variety of Nups reveals that both viruses proteolyze Nup98 and Nup153. Interestingly, ZIKA targets Tpr but not Nup62, while DENV targets Nup62 but not Tpr. More detailed analysis, again using Nup-specific antibodies reveals that these cleavage events result in loss of staining at the nuclear envelope with cleavage apparent by about 24 hpi, althouth Tpr and Nup153 cleavage in ZIKV-infected cells is apparent by 12 hpi. The addition of serine protease inhibitors prevents cleavage during DENV infection and expression of NS2B-NS3 via transient transfection recapitulates cleavage of Nups.

General Comments:

This study presents a large amount of data that supports the author’s conclusions that both ZIKV and DENV proteolyze NUPS via their NS2B-NS3 protease. These results are significant and convincing. The authors should consider adding additional statistical analyses to support their microscopy data such as the percentage of examined cells that display the phenotype presented in the figures.

Response

A statistical analysis in all microscopy data was included (Figures 1-7 and 9-10).

Another question is if the cleavage products produced by overexpression of the NS2B-NS3 protease mimic those found in infected cells.

Response

Under infection with ZIKV and DENV and after transfection of their NS2B-NS3 the cleavage products of Nup62, Nup98, and TPR are not detected. Thus, it is not possible to know if the same cleavage products are produced under both conditions. However, while the cleavage products of Nup153 are detected under infection, they were not observed after transfection. It is possible that under transfection conditions a higher amount of the protease is present and the cleavage/degradation of Nup153 is faster.

A minor point, but it isn’t clear why an analysis of Nup153 and Tpr in DENV-infected cells wasn’t undertaken, given that all the other virus/nup combinations were examined so thoroughly.

Response

As the reviewer clearly indicated, we did not include the analysis of Nup153 and TPR during DENV infection. TPR was not included because the infection with DENV does not induce cleavage/degradation of this Nup. In the case of Nup153, we could not do the experiments because we do not have enough amount of antibody to do the Western-blots to determine the Nup153 levels at different times post-infection with both DENV serotypes. Although we did not determine the integrity of Nup153 at different times post-infection with DENV, the experiment showed in figure 2 clearly indicate that Nup153 is cleavage/degraded at 24 hrs post-infection.

Lastly, it would be helpful for the reader who isn’t a flavivirus expert if the authors include some background on the time course of these infections. What do 12, 24, and 48 hpi correspond to in terms of the viral life cycle.

Response

A time course of the flaviviral replicative cycle was included (page 7 Line 246-253)

Specific Comments:

Fig1. Details on how the TEM was done should be added to the materials and methods and/or figure legend. This should include moi.

Response

The methodology used for cell infection and TEM was included in the materials and methods section (page 3, Line 124-133.

Fig 2. Statistical analysis should be added her showing the percentage of infected cells showing reduced 414 staining. Why the difference in time for microscopy vs western?

Response

A statistical analysis of the percentage of infected cells with reduced staining for Mab414 was included in Figure 2B.  We selected 48 hrs for confocal microscopy analysis to better observe the difference between infected and uninfected cells since, at this time, most of the cells in the preparation are infected.  In the case of the Western-blot assay, we tested the Nups integrity at 24 hpi because we wanted to detect the cleavage products using specific antibodies against each Nup, which was possible only for Nup153. 

Fig 3- how do the authors explain the lack of NE staining with the Nup62 ab? This is very different from what was seen with mAb414, which reacts with Nup62 and raises questions about the staining conditions and possible artifacts. The percent of cells showing loss of nuclear stain should be shown. If it’s similar to results obtained with mAb414, this will support the validity of the staining with this Ab.

Response

Mab414 recognizes Nup62 through its FG-rich sequence, which is also present in several other Nups. Thus, we used a polyclonal anti-Nup62 specific antibody to better determine NUP62 subcellular localization in both noninfected and infected cells. Moreover, using this anti-Nup62 specific antibody, the amount of the specific Nup in the cell was better shown. Additionally, it is known that besides the NE, many of the Nup are also present in the nucleoplasm, as has been widely reported.

The statistical analysis of the MFI obtained with both MAb414 and the specific anti- NUP62 antibodies is now included. A reduction of the MFI was obtained in infected cells stained with both antibodies when compared to the mock-infected cells supporting the validity of the staining with both antibodies.  

Figure 5,6- Again, statistics showing the % of cells showing the give pattern should be added.

Response

A statistical analysis of the infected cells showing a specific phenotype was included in both figures.

Fig 7- For Nup153 the loss of staining is not as apparent. Also some cells pos for viral E protein, seem to show normal staining of Nup153. Again, statistics may support the authors claims.

Response

A statistical analysis of the infected cells showing the specific phenotype was included.

Fig 9/10- statistics again are needed here to support the author’s interpretation

Response

A statistical analysis of the infected cells showing the specific phenotype was included.

Reviewer 2 Report

The authors are describing the role of viral nonstructural proteins in altering the integrity of NE and NPC. NS3 complexes with its cofactor NS2B to become NS2B3 complex that cleaves nucleoporins. The results are interesting and the research subject is novel. However, there are certain points need to be clarified.

Main comments

  1. In Figure 1, the authors claim that the viral NS2B3 complex induced changes in NE. By only using electron intensity, it is insufficient to tell what kind of change that has been done to the NE by NS2B3. I suggest that the authors should quantify the electron intensity and then compare between healthy cell and infected cells (both DENV- and ZIKV-infected cells). Furthermore, the group should show the complete NE in order to demonstrate the effect of NS2B3 on morphology of nucleus. It is interesting to investigate whether the components of NE such as lamin A/C or lamin B are the targets for NS2B3.
  2. In Figure 2B, the quality of western blot is poor. Please prepare data that are more convincing.
  3. In Figure 3, line 234, please add “(A)” in the figure legend.
  4. In Figure 5B, the western blot figure is poor; suggest getting a better one.
  5. In Figure 5, line 286, please change “the graph represent Nup62” to “…..represent Nup98”.
  6. In Figure 6A, please label “mock” and “ZIKV”. In line 294, please clarify which target is used to determine infectious status, E protein or NS3. Again in line 299, similar copy-paste mistake can be seen, please correct it to “The graph represents Nup98”.
  7. In Figure 8, the authors claim that NS2B3 cleaved nucleoporins and resulting degradation of the cleaved nucleoporins. However, protease inhibitors did not increase the protein level of the target nucleoporins. Furthermore, in Figure 11 and Figure 12, the cleavage products are not found in the western blot results. Therefore, the authors should perform qPCR to compare the mRNA levels of each target nucleoporins between uninfected cells and healthy cells to ensure that the reduction of the target nucleoporins are only at protein levels.
  8. Please standardize the NS2B-NS3 to NS2B3.
  9. Any explanation for the difference in selection of Nups for cleavage between DENV and ZIKV NS2B3 as both can cleave NUP153 and NUP98 but not NUP60/TPR.

Author Response

Reviewer 2

Thank you for your comments

Comments and Suggestions for Authors

The authors are describing the role of viral nonstructural proteins in altering the integrity of NE and NPC. NS3 complexes with its cofactor NS2B to become NS2B3 complex that cleaves nucleoporins. The results are interesting and the research subject is novel. However, there are certain points need to be clarified.

Main comments

  1. In Figure 1, the authors claim that the viral NS2B3 complex induced changes in NE. By only using electron intensity, it is insufficient to tell what kind of change that has been done to the NE by NS2B3. I suggest that the authors should quantify the electron intensity and then compare between healthy cell and infected cells (both DENV- and ZIKV-infected cells). Furthermore, the group should show the complete NE in order to demonstrate the effect of NS2B3 on morphology of nucleus. It is interesting to investigate whether the components of NE such as lamin A/C or lamin B are the targets for NS2B3.

Response

A statistical analysis as well as new figures were included showing the changes in the NE observed in DENV and ZIKV infected cells. Additionally, the interaction between viral proteins and lamin has been described previously. This information is now included in the discussion section. (Page 16 Lines 465-467)

  1. In Figure 2B, the quality of western blot is poor. Please prepare data that are more convincing.

Response

As the reviewer suggested, new representative Western-blot assays were included in Figure 2B and statistic analysis was also included in Figure 2C.

  1. In Figure 3, line 234, please add “(A)” in the figure legend.

Response

The “A” was added.

  1. In Figure 5B, the western blot figure is poor; suggest getting a better one.

Response

New Western-blot assay was added in Figure 5C

  1. In Figure 5, line 286, please change “the graph represent Nup62” to “…..represent Nup98”.

Response

Nup62 was changed for Nup98

  1. In Figure 6A, please label “mock” and “ZIKV”. In line 294, please clarify which target is used to determine infectious status, E protein or NS3. Again in line 299, similar copy-paste mistake can be seen, please correct it to “The graph represents Nup98”.

Response

In Figure 6A Mock and ZIKV infection is now indicated. The stain of NS3 was used to determine infectious status. This information is now included. “The graph represents Nup98” was corrected.

  1. In Figure 8, the authors claim that NS2B3 cleaved nucleoporins and resulting degradation of the cleaved nucleoporins. However, protease inhibitors did not increase the protein level of the target nucleoporins. Furthermore, in Figure 11 and Figure 12, the cleavage products are not found in the western blot results. Therefore, the authors should perform qPCR to compare the mRNA levels of each target nucleoporins between uninfected cells and healthy cells to ensure that the reduction of the target nucleoporins are only at protein levels.

Response

Although the qRT-PCR of the nucleoporins during infection could demonstrate if the reduction of the Nups levels is only at protein levels, there are two evidences presented in the manuscript that indicate that Nups are cleaved during DENV and ZIKV. The first, is the presence of the cleavage products of Nup153 in cells infected with DENV and ZIKV in Figure 2B. An important reduction in the uncleaved form of Nup153 is observed in infected cells compared with uninfected cells. The second evidence that the Nups are cleaved during viral infection is the presence of cleaved products of Nup98 and Nup62 in cells untreated with both serine-protease inhibitors compared with treated cells (Figure 8). This specific result indicates that the viral serine-protease or a virus induced serine-protease is involved in Nups degradation.

  1. Please standardize the NS2B-NS3 to NS2B3.

Response

NS2B3 was standardized throughout the manuscript

  1. Any explanation for the difference in selection of Nups for cleavage between DENV and ZIKV NS2B3 as both can cleave NUP153 and NUP98 but not NUP60/TPR.

Response

Differential target nucleoporins by the NS2B3 from DENV and ZIKV could be due to structural differences between both proteases. They only have 67% homology in their sequence. This type of differences for Nups cleavage has also been observed with different serotypes of Rhinovirus and also between different picornaviruses (Grifoni et al., 2017; Watters et al., 2017; Watters & Palmenberg, 2011). This information was included in the manuscript (Page 17 Line 487).

Reviewer 3 Report

The authors show that several nuclear pore proteins are relocated during DENV and ZIKV infections. In addition, there appears to be some specificity with regard to which nuclear pore proteins are relocated (for example NUP62 is not relocated by ZIKV). They show that relevant nups appear to be degraded by NS2B/NS3 protease and perhaps that the protease can relocate FG nups. They imply that the integrity of the nuclear membrane is disrupted. 

Major concerns:

The authors should show that the integrity of the nuclear membrane in another manner. EM is not sufficient. They need to show that cargo's that are not normally in the nucleus are present or vise versa, that transport out of the nucleus is altered. Is RNA transport altered? The demonstration of loss of nuclear membrane integrity by using  transport studies is important.

The authors show the NS2-3 protease degrades FG nups, and there is some indication that they also relocate, however the evidence is not strong. NS3 staining appears saturated in the images.  There should be accompanying quantification. In addition there should be an immuno-precipitation to show that there is an interaction between the FG nups and the protease. In fact active site mutants should still interact with the relevant nups, but it appears that mutant protease expression does not relocate FG nups. 

Minor comments:

The authors fail to reference work by the Tyrrell and Wozniak labs that shows HCV infection re-located Nups are to the membranous web . In addition, it was shown in these works that Nup 98 was re-located during DENV and HAV infections. They also fail to reference work by Lamarre which shows IMPB1 is cleaved by HCV NS3.

Author Response

Reviewer 3

Comments and Suggestions for Authors

The authors show that several nuclear pore proteins are relocated during DENV and ZIKV infections. In addition, there appears to be some specificity with regard to which nuclear pore proteins are relocated (for example NUP62 is not relocated by ZIKV). They show that relevant nups appear to be degraded by NS2B/NS3 protease and perhaps that the protease can relocate FG nups. They imply that the integrity of the nuclear membrane is disrupted. 

Major concerns:

The authors should show that the integrity of the nuclear membrane in another manner. EM is not sufficient. They need to show that cargo's that are not normally in the nucleus are present or vise versa, that transport out of the nucleus is altered. Is RNA transport altered? The demonstration of loss of nuclear membrane integrity by using transport studies is important.

Response

As the reviewer correctly indicated, the next step of this study is to analyze the effect of the modification in the NPC integrity in the transport of mRNAs, miRNAs and proteins. Since there are different nuclear-cytoplasmic pathways, it is necessary to analyze in further detail each one. Thus, the results should be part of another manuscript. However, it is well known that several nuclear proteins are relocated to cytoplasm during infection, suggesting that there are important modifications in the nuclear-cytoplasmic transport during infection. This information is described in the introduction section

The authors show the NS2-3 protease degrades FG nups, and there is some indication that they also relocate, however the evidence is not strong. NS3 staining appears saturated in the images.  There should be accompanying quantification. In addition there should be an immuno-precipitation to show that there is an interaction between the FG nups and the protease. In fact active site mutants should still interact with the relevant nups, but it appears that mutant protease expression does not relocate FG nups. 

Response

To support and validate our findings, quantification and statistical analysis in all experiments showed in all figures were included. On the other hand, although the presence of the wt form of NS2B3 from DENV and ZIKV reduces the integrity of the Nups analyzed, we do not know if the viral protease is directly inducing the cleavage of these Nups or if the viral protease is inducing the activation of a cellular serine-protease which could be responsible of cleaving the Nups. Both possibilities were included in the discussion section. Further studies should be performed to analyze the direct interaction between the Nups and the viral proteases as well as the affinity of this interaction, the specific sites cleaved by the proteases in each Nups and to analyze the differences between both proteases in the target Nups. 

Minor comments:

The authors fail to reference work by the Tyrrell and Wozniak labs that shows HCV infection re-located Nups are to the membranous web. In addition, it was shown in these works that Nup 98 was re-located during DENV and HAV infections. They also fail to reference work by Lamarre which shows IMPB1 is cleaved by HCV NS3.

Response

The references were included in line 482 and 530 respectively

Round 2

Reviewer 1 Report

The authors have conscientiously and thoroughly responded to all my original suggestions and queries. They have significantly improved the manuscript by adding statistical analyses of all the microscopy and TEM data along with providing an explanation of the time frame of the viral life cycle for non-experts and the protocols for the TEM analysis. The study represents a significant addition to our understanding of the interaction of Dengue and Zika the host nuclear pore complex.